# Wound complications following surgery to the lymph nodes: A protocol for a systematic review and meta-analysis

**Ikemsinachi C. Nzenwa**[1], **Hassan A. Iqbal**[1], **Claire Hardie**[2], **George E. Smith**[3], **Paolo L. Matteucci**[4], **Joshua P. Totty**[3,4,5]*

**1** Centre for Anatomical and Human Sciences, Hull York Medical School, Heslington, England, **2** Department of Plastic and Reconstructive Surgery, Leeds Teaching Hospitals NHS Trust, Leeds, United Kingdom, **3** Hull York Medical School, Heslington, England, **4** Department of Plastic and Reconstructive Surgery, Hull University Teaching Hospitals NHS Trust, Hull, England, **5** University of Hull, Hull, England

* Joshua.Totty@hyms.ac.uk

**Data Availability Statement:** No datasets were generated or analysed during the current study. All relevant data from this study will be made available upon study completion.

## Abstract

### Background

Malignancies that spread to the lymph nodes may be identified through surgical biopsy, and treatment of metastatic disease may be through lymph node dissection. These surgeries, however, may be associated with significant adverse outcomes, particularly wound complications, the true incidence of which remains unknown. Multiple studies have reported their individual rates of complications in isolation. The aim of this study will be to systematically evaluate data that presents the incidence of wound complications in patients undergoing these surgeries.

### Methods

We have designed and registered a protocol for a systematic review and meta-analysis of studies presenting incidence data. We will search MEDLINE, EMBASE and CENTRAL for relevant articles. Meta-analysis will be undertaken to synthesise an overall incidence of surgical site infection, wound dehiscence, haematoma and seroma. Subgroup analyses will investigate the effects of anatomical location, primary malignancy and study design on pooled incidence. Risk of bias will be evaluated for each included study using bespoke tools matched to the study design.

### Discussion

The results of this study will provide the incidence of wound complications and secondary complications following lymph node surgery. This will directly impact upon the consent process, and may influence the nature of future research studies aimed at reducing post-operative complications.

**Funding:** No direct funding was secured for this protocol or subsequent review. Joshua Totty is a Clinical Lecturer funded by Health Education England (HEE) / National Institute for Health Research (NIHR). The views expressed in this publication are those of the author(s) and not necessarily those of the NIHR, NHS or the UK Department of Health and Social Care.

**Competing interests:** The authors have declared that no competing interests exist.

# Background

Sentinel lymph node biopsy (SLNB) was developed in the early 1990s and became the standard diagnostic tool for evaluating metastasis to the lymph node and staging malignancy [1]. A positive SLNB, indicating the spread of malignancy to the regional nodes, may indicate the need for axillary (ALND) or inguinal (ILND) block dissection, where the entire lymph node basin is removed [1,2]. These complete dissections continue to be a staple in managing metastatic disease, even though their efficacy is still heavily debated [1–3].

Block dissection may be indicated for malignancy of the breast, malignant melanoma and other cutaneous malignancies, and urogenital cancers. However, a number of factors dispose these surgeries to an increased rate of post-operative complications, which may result in a reduced quality of life [2]. Wound complications constitute some of the major adverse outcomes following lymph node surgery, regardless of the site of the dissection. These complications are not limited to surgical site infections and also include wound dehiscence, delayed wound healing, seroma and hematoma [4].

A recent case series of 244 ILNDs reported at least one wound complication in 51.2% (n = 124) of participants. 29.8%, 21.5% and 5% of the total population developed wound infections, seroma and hematomas, respectively, and irrespective of patient and operative factors [3].

A myriad of interventions for reducing rates of wound complications following surgery have been described in the literature and have been adopted to varying degrees. To assess the potential impact of these interventions, an estimate of the true incidence of complications following surgery to the lymph nodes is required.

## Study objectives

The aim of this study is to identify the incidence of wound complications following biopsy and dissection of the axillary or inguinal lymph nodes, and further identify the rate of secondary complications. The PICO framework for this review is as follows:

Population–any adult undergoing surgery to the lymph nodes in the axilla or groin for the purposes of diagnosing or treating cancer.

Intervention and Control–As this is a study of proportions, no specific intervention or control are being examined. The procedures of interest are lymph node biopsy or completion lymphadenectomy.

Outcome–The outcome of interest is one of four wound complications, namely surgical site infection, seroma, haematoma or wound dehiscence.

# Methods

## Systematic review registration

The protocol has been registered with PROSPERO International Prospective Register of Systematic Reviews (registration number CRD42021239530). This protocol is reported in line with the Preferred Reporting Items for Systematic Reviews and Meta-Analysis Protocols (PRISMA-P) (See *S1 Table*) [5]. This review will be conducted in accordance with the Cochrane Handbook for Systematic Review of Interventions [6].

## Search strategy and information sources

MEDLINE, EMBASE and CENTRAL are the primary literature sources searched for original studies published, in English language, from inception to the date searches are conducted. The search strings to be used for the database searches have been included in the *S2 Table* and has

been designed in conjunction with an information search specialist. Additionally, the reference lists of all included studies will be searched manually to identify any additional studies that fulfil the inclusion criteria.

## Inclusion and exclusion criteria

Randomized and non-randomized studies that report wound complications (wound infections, wound dehiscence, seroma and hematoma) following axillary or inguinal lymph node biopsy or dissection for adult patients (age ≥18) with malignancy will be included. The diagnostic, therapeutic or prophylactic nature of the LND will not be taken into account when including the studies.

Case series with a sample size of less than 20 patients will be excluded. All commentaries, case reports, conference abstracts, literature overviews, literature reviews and meta-analyses, secondary analyses for previously published articles and non-English studies will be excluded. However the reference lists of relevant review articles will be hand-searched for articles missing from the original searches.

## Study selection

Three stages have been outlined for the study selection. Search results will be uploaded to Rayyan [7], a bespoke web and mobile app for conducting systematic reviews, and titles and abstracts screened by two reviewers acting independently. Following this, full-text articles will be retrieved and examined, and assessed against the inclusion/exclusion criteria by the same two reviewers. Finally, the additional studies from the references of selected studies will be screened. Any discrepancy between reviewers will be resolved by discussion with a third reviewer, and any further discrepancy or disagreement will be put to consensus between all authors. Study inclusion and exclusion, and the reasons for exclusion at the full-text stage, will be reported using a standardised PRISMA flow-diagram [8].

## Primary and secondary outcomes

The primary outcome is the incidence of wound complications at 30 days, defined as infection, dehiscence, seroma or hematoma following surgery to the axillary or inguinal lymph nodes. Where data allows, each complication will be examined individually. Where the data is available, secondary outcomes will include the use of postoperative antibiotics within 30 postoperative days (not including antibiotic prophylaxis), the incidence of return to theatre for postoperative complications within 30 days and 30-day mortality.

## Study designs

It is anticipated that the searches will return studies that are either randomised intervention studies, non-randomised intervention studies, and observational/cross sectional studies reporting incidence data. In order to achieve the best possible estimate, all three study types will be considered for analysis. Where a study has multiple arms (such as an interventional study), each arm will be considered within the analysis independently.

## Data extraction

Three reviewers will extract the data from the full-text articles independently, using a standardized electronic data extraction sheet designed for this systematic review. The extraction sheet will be designed in Microsoft Excel and any disagreements will be resolved by discussion with the third reviewer.

Where possible, the following information will be extracted from the individual papers: *study characteristics* (first author, publication year, country where data was collected from, study design, journal), *patient characteristics* (number of patients included, number of males and females, mean or median age with standard deviation, comorbidities, site of SLNB or LND), *postoperative outcomes* (number of wound infections, wound dehiscence, seroma, hematoma, mortality) and *secondary interventions* (number of reoperations for any wound complication, antibiotic use, and 30-day mortality).

If there is missing data, the respective authors will be contacted if the email address has been provided in the article.

## Assessment of risk of bias of included studies

The risk of bias will be assessed at the study level by two reviewers independently, and discrepancies will be resolved in the same fashion as used for study inclusion/exclusion. The risk of bias tool used will be a risk of bias tool specifically designed for prevalence studies [9].

## Data analysis and synthesis

The extracted data will be pooled and combined statistically to assess the primary and secondary outcomes. For the primary outcome of the study, wound complications will be presented as a crude incidence estimate expressed in percentages with 95% confidence intervals, for each individual complication (total four analyses). We will use a generalised linear mixed model (GLMM) to synthesis proportions and present the results with a forest plot. GLMMs are considered to be superior to traditional two step methods with either logit or double arcsine transformations [10,11]. As a sensitivity analysis, a random effects meta-analysis using the Freeman-Tukey double arcsine transformation will also be undertaken and presented as supplementary material. Heterogeneity will be evaluated using the $I^2$ statistic test, and interpretation will be according to that provided in the *Cochrane Handbook* [6]. An $I^2$ statistic of over 75% (indicating considerable heterogeneity) will prompt us to explore the heterogeneity by way of meta-analyses by subgroup.

A funnel plot will be produced to examine the effect of publication bias, or small study effects, if more than 10 studies are included in the analysis [12].

## Additional analyses

Where possible, separate subgroup analysis will be undertaken to assess the effects of important potential confounding factors upon the primary outcomes, including study design, the nature of the primary malignancy (skin, breast, urogenital, etc), the effect of the site of procedure (axillary vs inguinal surgery) and the effect of the type of procedure (biopsy vs block dissection). No meta-regression analyses are planned at this stage, as we anticipate significant statistical and clinical heterogeneity between groups.

## Discussion

Surgery to the lymph nodes is common, and as worldwide rates of malignancy increase, it is likely the frequency at which these procedures are undertaken also will rise. This review will aim to identify the incidence of significant complications, which will inform the consent and decision-making process on an individual level, and may be used to direct future research studies to improve outcomes on a population level.

In this paper we have outlined a protocol for a systematic review and meta-analysis, in line with current best-practice guidelines that recommend the pre-publication of protocols such as

this. Any deviations from this protocol will be reported in the final manuscript, and search strategies, results, data extraction and analyses will be held in an open-access repository. Dissemination will be through presentation at national meetings/conferences, and through robust peer-reviewed publication.

We anticipate limitations in the review, in the form of heterogenous data and a wide variance in reported rates of complications. Data on surgical site infection is difficult to collect, and there is no consensus on the definition of outcomes such as surgical site infection [13]. We aim to overcome this by placing no restrictions on definitions used, and taking the data presented in individual studies as being valid.

In summary, we have presented the protocol for a wide-ranging systematic review that aims to provide further insight into the incidence of significant complications following common surgical procedures.

## Supporting information

**S1 Table. PRISMA-P 2015 checklist for 'wound complications following surgery to the lymph nodes: A protocol for a systematic review and meta-analysis'.**
(DOCX)

**S2 Table. Search strategy for 'wound complications following surgery to the lymph nodes: A protocol for a systematic review and meta-analysis'.**
(DOCX)

## Acknowledgments

JPT is the guarantor of the study. JPT conceived, conceptualized and planned the protocol of the study. ICN registered the protocol and wrote the first draft of the manuscript. All authors provided intellectual input into critical revisions of the protocol and the final version is the collective effort of all six authors. We would also like to thank Tim Staniland, of Hull University Teaching Hospitals NHS Trust Library, for his assistance with developing the search strategy.

## Author Contributions

**Conceptualization:** Paolo L. Matteucci, Joshua P. Totty.

**Methodology:** Ikemsinachi C. Nzenwa, Hassan A. Iqbal, Claire Hardie, George E. Smith, Joshua P. Totty.

**Supervision:** Claire Hardie, George E. Smith, Paolo L. Matteucci, Joshua P. Totty.

**Writing – original draft:** Ikemsinachi C. Nzenwa.

**Writing – review & editing:** Ikemsinachi C. Nzenwa, Hassan A. Iqbal, Claire Hardie, George E. Smith, Paolo L. Matteucci, Joshua P. Totty.

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
