## [Decision Letter · Decision Letter 0]

30 Jun 2021

PONE-D-21-12899

Wound Complications Following Surgery to the Lymph Nodes: A protocol for a systematic review and meta-analysis

PLOS ONE

Dear Dr. Joshua P Totty,

Thank you for submitting your manuscript to PLOS ONE. After careful consideration, we feel that it has merit but does not fully meet PLOS ONE’s publication criteria as it currently stands. Therefore, we invite you to submit a revised version of the manuscript that addresses the points raised during the review process.

We look forward to receiving your revised manuscript.

Kind regards,

Paolo Aurello

Academic Editor

PLOS ONE

Journal Requirements:

Reviewers' comments:

Reviewer's Responses to Questions

**Comments to the Author**

1. Does the manuscript provide a valid rationale for the proposed study, with clearly identified and justified research questions?

Reviewer #1: Yes

2. Is the protocol technically sound and planned in a manner that will lead to a meaningful outcome and allow testing the stated hypotheses?

Reviewer #1: Yes

3. Is the methodology feasible and described in sufficient detail to allow the work to be replicable?

Reviewer #1: Yes

4. Have the authors described where all data underlying the findings will be made available when the study is complete?

Reviewer #1: Yes

5. Is the manuscript presented in an intelligible fashion and written in standard English?

Reviewer #1: Yes

6. Review Comments to the Author

You may also provide optional suggestions and comments to authors that they might find helpful in planning their study.

Reviewer #1: Thank you for the opportunity to review this article. It is a well planned and structured meta-analysis. However authors declerated that results will be undertaken after the registration of the protocol on PROSPERO. Evaluating an article without the Results section is quite confusing. It would be appropriate to re-evaluate after the process and the article are completed.

7. PLOS authors have the option to publish the peer review history of their article (what does this mean?). If published, this will include your full peer review and any attached files.

Reviewer #1: No

---

## [Author Response · Author response to Decision Letter 0]

19 Jul 2021

13th July 2021

Dear Editors,

Thank you for your review of our manuscript, Wound Complications Following Surgery to the Lymph Nodes: A protocol for a systematic review and meta-analysis. 

We were confused by the content of the review. In the review, the article is deemed to meet the criteria for publication (The reviewer’s responses to all questions is “Yes”). It clearly states that the manuscript is a protocol, and was submitted as such during the submission process. However, the reviewer has refused to accept the article, citing “Evaluating an article without the Results section is quite confusing. It would be appropriate to re-evaluate after the process and the article are completed.” 

We feel the reviewer has not understood that the article is a review protocol, rather mistaking it for a half completed systematic review. As such, we have no substantial changes to make based on the reviewers comments.

Please find our responses to each individual point:

Editors Comments:

Amended and updated. Thanks.

Data availability applies to the completed review. As this submission is a study protocol, no data will be generated and therefore the submission should not be withheld. Full data in a repository will be provided with the completed review in a separate submission. Thanks

Amended and Updated. Thanks.

Reviewer’s Comments:

Thank you for the opportunity to review this article. It is a well planned and structured meta-analysis. However authors declerated that results will be undertaken after the registration of the protocol on PROSPERO. Evaluating an article without the Results section is quite confusing. It would be appropriate to re-evaluate after the process and the article are completed.

Thank you for your comments. The protocol is registered on PROSPERO and the registration information is contained within the manuscript. This is a study protocol and as such no results have been generated. Publishing protocols is deemed good practice and is practiced by the Cochrane Collaboration. Results will be available in due course once the study has been completed, and will be submitted to an appropriate peer reviewed journal.

As per your email of 14th July, please amend the funding statement to “The authors received no specific funding for this work.”

We have resubmitted the article with some formatting changes as requested by the journal. We fail to understand, based on this review, why our article would be rejected for PLOS ONE: Protocols, when there was no significant criticism of the protocol we presented. We hope to receive your response in a timely fashion.

Kind regards,

Joshua Totty

MBBS, PGCertRes, MRCS, MD, FHEA

Registrar in Plastic and Reconstructive Surgery (ST3) - Hull University Teaching Hospitals NHS Trust

Yorkshire and Humber Deanery

NIHR Clinical Lecturer in Plastic Surgery

Hull York Medical School

---

## [Editor Report · Decision Letter 1]

2 Sep 2021

PONE-D-21-12899R1

Wound Complications Following Surgery to the Lymph Nodes: A protocol for a systematic review and meta-analysis

PLOS ONE

Dear Dr. Joshua P Totty

Thank you for submitting your manuscript to PLOS ONE. After careful consideration, we have decided that your manuscript does not meet our criteria for publication and must therefore be rejected.

I am sorry that we cannot be more positive on this occasion, but hope that you appreciate the reasons for this decision.

Yours sincerely,

Paolo Aurello

Academic Editor

PLOS ONE

---

## [Author Response · Author response to Decision Letter 1]

20 Sep 2021

Dear Editors,

As part of the appeal process, we are required to submit a response to reviewers. The entire content of the review is as follows:

“Thank you for submitting your manuscript to PLOS ONE. After careful consideration, we have decided that your manuscript does not meet our criteria for publication and must therefore be rejected.

I am sorry that we cannot be more positive on this occasion, but hope that you appreciate the reasons for this decision.”

As you may see, there is little feedback given for us, as authors to respond to.

In the reply, you cite that the manuscript does not meet publication criteria. You do not state how, or why, it does not meet these criteria.

• The manuscript is a study protocol, which PLOS ONE advertises as publishing - see https://plos.org/protocols/

• Indeed, it is the protocol for a systematic review, which PLOS ONE does publish, as evidenced by the following articles published this month:

o https://doi.org/10.1371/journal.pone.0256596

https://doi.org/10.1371/journal.pone.0255789

• Furthermore, when the submission originally went for peer review, it was deemed to meet the criteria for publication by the reviewer (See below):

1. Does the manuscript provide a valid rationale for the proposed study, with clearly identified and justified research questions?

Reviewer #1: Yes

2. Is the protocol technically sound and planned in a manner that will lead to a meaningful outcome and allow testing the stated hypotheses?

Reviewer #1: Yes

3. Is the methodology feasible and described in sufficient detail to allow the work to be replicable?

Reviewer #1: Yes

4. Have the authors described where all data underlying the findings will be made available when the study is complete?

Reviewer #1: Yes

5. Is the manuscript presented in an intelligible fashion and written in standard English?

Reviewer #1: Yes

As authors we responded to the review positively and resubmitted in a timely fashion.

In summary, I do not "appreciate the reasons for this decision” and we as authors are left confused by the fact that, despite minimal changes between revisions, an article can be deemed unpublishable after 4 months of submission.

We look forward to the outcome of the appeal.

Kind regards,

Joshua Totty

MBBS, PGCertRes, MRCS, MD, FHEA

Registrar in Plastic and Reconstructive Surgery (ST3) - Hull University Teaching Hospitals NHS Trust

Yorkshire and Humber Deanery

NIHR Clinical Lecturer in Plastic Surgery

Hull York Medical School

---

## [Decision Letter · Decision Letter 2]

4 May 2022

PONE-D-21-12899R2Wound Complications Following Surgery to the Lymph Nodes: A protocol for a systematic review and meta-analysisPLOS ONE

Dear Dr. Totty,

Thank you for submitting your manuscript to PLOS ONE. After careful consideration, we feel that it has merit but does not fully meet PLOS ONE’s publication criteria as it currently stands. Therefore, we invite you to submit a revised version of the manuscript that addresses the points raised during the review process.

The mansucript has been reviewed by one reviewer and their comments may be seen below. The reviewer believes that the aim of the systematic review as well as the review objectives and research question  could be clarified.

Could you please revise the manuscript to carefully address the concerns raised?

We look forward to receiving your revised manuscript.

Kind regards,

Lucinda Shen, MSc

Staff Editor

PLOS ONE

Journal Requirements:

1. Please note that in order to use the direct billing option the corresponding author must be affiliated with the chosen institute. Please either amend your manuscript to change the affiliation or corresponding author, or email us at plosone@plos.org with a request to remove this option.

“No direct funding was secured for this protocol or subsequent review. Joshua Totty is a Clinical Lecturer funded by Health Education England (HEE) / National Institute for Health Research (NIHR). The views expressed in this publication are those of the author(s) and not necessarily those of the NIHR, NHS or the UK Department of Health and Social Care.”

We note that you have provided funding information within the funding Section that is not currently declared in your Funding Statement. Please note that funding information should not appear in the Acknowledgments section or other areas of your manuscript. We will only publish funding information present in the Funding Statement section of the online submission form.

“The authors received no specific funding for this work.”

Additional Editor Comments (if provided):

Reviewers' comments:

Reviewer's Responses to Questions

**Comments to the Author**

1. Does the manuscript provide a valid rationale for the proposed study, with clearly identified and justified research questions?

Reviewer #2: Yes

2. Is the protocol technically sound and planned in a manner that will lead to a meaningful outcome and allow testing the stated hypotheses?

Reviewer #2: Yes

3. Is the methodology feasible and described in sufficient detail to allow the work to be replicable?

Reviewer #2: Yes

4. Have the authors described where all data underlying the findings will be made available when the study is complete?

Reviewer #2: Yes

5. Is the manuscript presented in an intelligible fashion and written in standard English?

Reviewer #2: Yes

6. Review Comments to the Author

You may also provide optional suggestions and comments to authors that they might find helpful in planning their study.

Reviewer #2: The protocol is well presented with appropriate details. Authors have mentioned the aim of the study. It would be more clearer if there is any mention on the study objectives and what are the specific review questions. These could be spelt clearly before describing the methods. Systematic reviews and protocols generally have the eligibility criteria mentioned in the PICO framework (also mentioned in the PRISMA-P). The authors need to mention this in the protocol.

7. PLOS authors have the option to publish the peer review history of their article (what does this mean?). If published, this will include your full peer review and any attached files.

Reviewer #2: **Yes: **

---

## [Author Response · Author response to Decision Letter 2]

12 May 2022

Reply to reviewers:

1. Please note that in order to use the direct billing option the corresponding author must be affiliated with the chosen institute. Please either amend your manuscript to change the affiliation or corresponding author, or email us at plosone@plos.org with a request to remove this option.

Thank you. I can confirm that as the corresponding author I am affiliated with the University of Hull, listed in my affiliations in the manuscript.

“No direct funding was secured for this protocol or subsequent review. Joshua Totty is a Clinical Lecturer funded by Health Education England (HEE) / National Institute for Health Research (NIHR). The views expressed in this publication are those of the author(s) and not necessarily those of the NIHR, NHS or the UK Department of Health and Social Care.”

We note that you have provided funding information within the funding Section that is not currently declared in your Funding Statement. Please note that funding information should not appear in the Acknowledgments section or other areas of your manuscript. We will only publish funding information present in the Funding Statement section of the online submission form.

“The authors received no specific funding for this work.”

Thank you. The funding statement should read “No direct funding was secured for this protocol or subsequent review. Joshua Totty is a Clinical Lecturer funded by Health Education England (HEE) / National Institute for Health Research (NIHR). The views expressed in this publication are those of the author(s) and not necessarily those of the NIHR, NHS or the UK Department of Health and Social Care”

Thank you. I can confirm the reference list is up to date.

Reviewer #2: The protocol is well presented with appropriate details. Authors have mentioned the aim of the study. It would be more clearer if there is any mention on the study objectives and what are the specific review questions. These could be spelt clearly before describing the methods. Systematic reviews and protocols generally have the eligibility criteria mentioned in the PICO framework (also mentioned in the PRISMA-P). The authors need to mention this in the protocol.

We thank the reviewer for their comments. We have updated and amended the manuscript as suggested, with the addition of a section titled “study objectives”, including a PICO framework, before the methods section.

---

## [Decision Letter · Decision Letter 3]

21 Jul 2022

Wound Complications Following Surgery to the Lymph Nodes: A protocol for a systematic review and meta-analysis

PONE-D-21-12899R3

Dear Dr. Totty,

We’re pleased to inform you that your manuscript has been judged scientifically suitable for publication and will be formally accepted for publication once it meets all outstanding technical requirements.

Kind regards,

Carla Pegoraro

Division Editor

PLOS ONE

Additional Editor Comments (optional):

Reviewers' comments:

Reviewer's Responses to Questions

**Comments to the Author**

1. Does the manuscript provide a valid rationale for the proposed study, with clearly identified and justified research questions?

Reviewer #2: Yes

2. Is the protocol technically sound and planned in a manner that will lead to a meaningful outcome and allow testing the stated hypotheses?

Reviewer #2: Yes

3. Is the methodology feasible and described in sufficient detail to allow the work to be replicable?

Reviewer #2: Yes

4. Have the authors described where all data underlying the findings will be made available when the study is complete?

Reviewer #2: Yes

5. Is the manuscript presented in an intelligible fashion and written in standard English?

Reviewer #2: Yes

6. Review Comments to the Author

You may also provide optional suggestions and comments to authors that they might find helpful in planning their study.

Reviewer #2: The protocol is presented well. The population included is still broad without any specific category. All adults, with any primary site malignancy, any stage, any duration of malignancy, any co-morbid conditions, any cancer treatment group etc. Authors need to bear this in mind when you pool the data and genenralise the outcome. Sub-group analysis will be very valuable. Lines 124, 125 mentions the primary outcomes as wound complications at 30 days. This 30 days criteria has not come up anywhere in the background and methodology. The word haematoma is mentioned as hematoma at several places. Need to have uniformity. Lines 109 and 110 are not relevant. Data extraction template could have been presented as appendix. The search terms, statistical analysis proposed, PRISMA check list is appropriate.

7. PLOS authors have the option to publish the peer review history of their article (what does this mean?). If published, this will include your full peer review and any attached files.

Reviewer #2: **Yes: **Arun Kumar Basavaraj

---

## [Editor Report · Acceptance letter]

26 Jul 2022

PONE-D-21-12899R3 

Wound Complications Following Surgery to the Lymph Nodes: A protocol for a systematic review and meta-analysis 

Dear Dr. Totty:

I'm pleased to inform you that your manuscript has been deemed suitable for publication in PLOS ONE. Congratulations! Your manuscript is now with our production department. 

Kind regards, 

on behalf of

Dr Carla Pegoraro 

Staff Editor

PLOS ONE